# Clinical Application of Monitoring Vital Signs in Dogs Through Ballistocardiography (BCG)

**DOI:** 10.3390/vetsci12040301

**Published:** 2025-03-24

**Authors:** Bolortuya Chuluunbaatar, YungAn Sun, Kyerim Chang, HoYoung Kwak, Jinwook Chang, WooJin Song, YoungMin Yun

**Affiliations:** 1College of Veterinary Medicine and Veterinary Medical Research Institute, Jeju National University, Jeju 63243, Republic of Korea; ch.loboroo@gmail.com (B.C.); quincy0825@naver.com (Y.S.); ssong@jejunu.ac.kr (W.S.); 2CareSix, Jeju International University, Jeju 63309, Republic of Korea; krchang@cotons.ai; 3Department of Computer Engineering, Jeju National University, Jeju 63243, Republic of Korea; kwak@jejunu.ac.kr; 4HRG, Jeju International University, Jeju 63309, Republic of Korea; kerimc14@gmail.com

**Keywords:** electrocardiography (ECG), ballistocardiography (BCG), wearable device, heart rate, respiratory rate, anesthesia

## Abstract

This study explored a new way to monitor the heart and breathing rates of dogs using a wearable device based on the Ballistocardiography (BCG) method. Compared to the traditional method using electrocardiography (ECG), which is accurate but requires shaving the animal and attaching electrodes, which could be time-consuming and stressful, this device is non-invasive, easy to use, and allows for continuous monitoring without extensive preparation. This device was tested on both normal dogs and anesthetized dogs to check its efficiency in comparison with electrocardiography (ECG). The results showed that under normal conditions, the wearable device gives reliable data, and it is efficient at detecting heart and breathing rates. However, under anesthesia, when irregular heartbeats might occur, the device would often give divergent readings. Despite this drawback, BCG proved useful because of its rapidity and non-invasive nature; hence, it presented itself as a very helpful tool for veterinarians. This technique could save time and reduce stress in animals and, therefore, be a practicable solution for routine and surgical monitoring in veterinary medicine.

## 1. Introduction

As the number of households with companion animals increases, interest in health management and the early diagnosis of diseases is also increasing [1,2]. In both human medicine and veterinary medicine, measuring vital signs such as body temperature (BT), heart rate (HR), and respiration rate (RR) is an essential diagnostic item for assessing health status [3]. These vital signs provide important information about the health status of animals and help us respond quickly in emergency situations [4].

The electrocardiogram (ECG) and photoplethysmography (PPG) methods, which are generally used to evaluate cardiopulmonary function, can cause some discomfort and stress to animals [5,6]. In order to measure with these methods, hair must be shaved, and sensors or patches must be directly attached to the skin [7]. The ECG and PPG equipment currently used for animals are modified versions of human equipment and may not be suitable depending on the anatomical differences in animals, their various sizes (from small to extra-large), the presence or absence of hair, and hair density. To solve these problems, intelligent wearable monitoring systems that can minimize stress and increase the accuracy and efficiency of cardiopulmonary function evaluation are needed; vital sign monitoring systems using MEMS microphones and alternative measurement systems that can record cardiac activity are being developed [8,9].

Ballistocardiography (BCG) is a noninvasive diagnostic method first discovered by Gordon in 1877 [10]. It graphically displays the microscopic movements of the body caused by the physical elasticity of blood vessels as blood is pumped out through the contraction and relaxation of the heart [11,12]. This graph indirectly reflects the activity of the heart, and a normal BCG shows a consistent heartbeat and consists of seven waveforms (H, I, J, K, L, M, and N waves) [13,14]. The BCG waveform includes various biological information such as heartbeat interval (instantaneous pulse rate), heart rate variability (HRV) characteristics, cardiac contractility, and blood pressure (BP) [15,16]. In veterinary research, there are some papers on the health status assessment of horses, cattle, and sheep, but research on small animals such as dogs and cats is relatively lacking [17,18]. There is a need to develop equipment that can accurately and periodically monitor the vital signs of animals while minimizing stress. This study aims to evaluate the clinical utility of a BCG wearable device (Coton Sense1) developed to monitor vital signs in dogs.

## 2. Materials and Methods

### 2.1. Subjects

This study was carried out at the Jeju National University Veterinary Teaching Hospital from July 2023 to November 2024 on a group of 12 dog subjects. The subjects were categorized into two groups: 6 dogs (client-owned) undergoing neutering and spaying surgery, representing the anesthetized cohort, and 6 beagles (purpose-owned) representing the awake cohort (Table 1). This enabled the comparison of the physiological parameters of the anesthetized and awake canine subjects. They were tested with blood tests (ProCyte Dx, IDEXX Laboratories, Inc., Westbrook, ME, USA), serum biochemical tests (Catalyst One analyzer, IDEXX Laboratories, Inc, Westbrook, ME, USA), PCR tests for tick-borne diseases (Babesiosis, Hepatozoonosis), and heartworm diagnostic kits (HW Rapid kit, BioNote, Hwaseong-si, Republic of Korea). These evaluations confirmed that both groups of dogs were free from hematologic and biochemical abnormalities and infections and were not on any medications.

### 2.2. Data Collection

For this study, a comprehensive dataset comprising 10,000 raw data points was collected for both respiratory rate (RR) and heart rate (HR) readings in the 2 dog groups. Following the application of stringent data filtering protocols to ensure accuracy and reliability, 8000 high-quality data points were retained for subsequent analysis. This refined dataset provided a good comparison between HR and RR measures, enabling close scrutiny of the physiological parameters under investigation. For the awake dog group, 6 awake beagles (3 male and 3 females; mean age: 3.5 years and range: 3.0–3.8 years; mean weight: 12.4 kg and range: 11.2–14.1 kg) were used. Resting and post-exercise ECG and BCG recordings were obtained after a standardized 30 min running exercise protocol, while in the anesthetized group, 6 client-owned dogs (4 males and 2 females; mean age: 4.6 years and range: 1.0–8.2 years; mean weight: 24.2 kg and range: 19.4–31.2 kg) were used.

### 2.3. Instrumentation and Measurement Techniques 

In this study, HR and RR measurements with a duration of 30 min to 1 h were obtained using ECG and BCG devices. These measurements were carried out under various conditions, such as at rest, after exercise, and under anesthesia, to enable extensive data collection across different physiological conditions.

#### 2.3.1. Ballistocardiography (BCG)

The wearable BCG device (Coton Sense1, CareSix, Republic of Korea) monitors vital signs continuously, 24 h a day. It integrates a piezoelectric sensor, a 6-axis sensor (ICM-42605 San Jose, CA, USA), microelectromechanical technology (ICM-40619 (USA)), and a temperature and humidity sensor (CMOSense chipCareSix, Seoul, Republic of Korea). Data collected include vital signs and movement information, which are transmitted to the Cotons Vet Cloud server via the Gateway.

#### 2.3.2. Electrocardiogram (ECG)

Routine electrocardiogram recordings were made using a patient monitor (BM3Vet Pro, Bionet, Republic of Korea). The electrocardiogram is a standard method for measuring the electrical activity of the heart and was used as a reference for verifying BCG measurements. To obtain an ECG using a patient monitor, the hairs of the animals’ left and right anterior and left hindlimb proximal areas were shaved; electrode patches were attached, and the electrodes were connected. The ECG for comparison with BCG was made using the second lead.

#### 2.3.3. Heart Rate (HR) Measurement

HR data were recorded from both BCG and ECG instruments so that cardiac activity could be measured continuously. HR was also manually measured using two techniques: femoral artery palpation and stethoscope auscultation. The index and middle fingers for palpation of the femoral pulse were positioned on the inner aspect of the thigh along the course of the femoral artery to feel for pulsations. For auscultation to hear heartbeats, place the stethoscope on the left chest slightly posterior to the elbow. The hand techniques were secondary checks for verifying HR data collected using BCG and ECG equipment.

#### 2.3.4. Respiratory Rate (RR) Measurement

RR was ascertained through direct visual monitoring of respiratory movements, a method that is widely considered the gold standard in clinical veterinary practice because of its simplicity, reliability, and direct assessment of respiratory mechanics. While ECG and BCG devices may display respiratory information, direct visual monitoring served as the reference standard in the scenario of this study. Furthermore, the Bionet BM3Vet Pro veterinary monitor used in this investigation has the feature of capnography and provides the capability for mainstream and side stream monitoring of end-tidal CO_2_ (EtCO_2_). By employing capnography, we were able to derive continuous and unbiased information regarding respiratory function, and hence, the evaluation of anesthesia effectiveness of ventilation was improved.

#### 2.3.5. Anesthetic Procedure and Positioning

All dogs were positioned in dorsal recumbency, a standard posture for abdominal operations including spaying and castration, during anesthesia since it provides the best access to the surgical site and helps to monitor vital signs effectively. Continuous HR and RR measurements were obtained using a gold standard ECG and BCG wearable device throughout the anesthesia procedure under isoflurane anesthesia, with each measurement lasting between 30 min and 1 h. This comprehensive monitoring ensured the accurate assessment of cardiovascular and respiratory parameters throughout the perioperative period.

### 2.4. Statistical Analysis

In both groups, the agreement and correlation between HR and RR obtained from the ECG through conventional patient monitoring and BCG measured using wearable device were analyzed using the Bland–Altman plot and linear regression method.

#### 2.4.1. Bland–Altman Plot

This method assesses the agreement between two quantitative measurements by plotting the differences against their averages, allowing for the identification of systematic biases and the estimation of limits of agreement. The methodology is especially useful in clinical studies intended to demonstrate the interchangeability of measurement methods.

#### 2.4.2. Linear Regression Analysis

The correlation between HR and RR measurements obtained from the wearable device and the reference ECG machine was analyzed using linear regression. This analysis method evaluated the strength and direction of the relationship between the two measurement methods by applying a regression line to the data. 

## 3. Results

### 3.1. Awake Beagle Group

Our results strongly suggest that BCG is a powerful new technology that opens the potential for a significant improvement of veterinary monitoring capabilities as a non-invasive, accurate, and efficient alternative to ECG. Both BCG and ECG were measured for HR and RR monitoring, and studies have shown high agreement between these methods. For instance, this research has demonstrated that BCG signals correlate strongly with ECG measurements, particularly in the J and R waves, suggesting that BCG can effectively reflect cardiac activity. As shown in (Figure 1). ECG and BCG measurements of HR and RR showed high agreement, with strong correlations (HR: r = 0.97, *p* < 0.001; RR: r = 0.78, and *p* < 0.001). Regression models indicated strong predictive relationships for both HR (R^2^ = 0.94) and RR (R^2^ = 0.61).

Figure 1 Comparison of filtered BCG signals with ECG signals in awake dogs during (HR) monitoring.

#### 3.1.1. Heart Rate

The Bland–Altman analysis showed a mean bias of −0.212 bpm, with limits of agreement (LOA) between −5.917 and 5.492 bpm. This suggests a minor HR overestimation of the BCG Sense1 wearable device in comparison with ECG, though it remains within an acceptable range, validating its low error of measurement in conscious animals, while linear regression analysis (BCG = 1.03 × ECG-2.4108) demonstrated a close correlation (R^2^ = 0.9399), confirming the high accuracy of the device for monitoring the heart rates of awake dog As shown in (Figure 2).

Figure 2 Comparison of HR between the ECG and BCG in awake dogs.

#### 3.1.2. Respiratory Rate

Respiratory rates from BCG and ECG recordings acquired in awake dogs had minimal bias (0.35 bpm) but substantial variability (LOA: −4.2 to 4.9 bpm), demonstrating evidence of acceptable average agreement but poor individual concordance, whereas the analysis with linear regression showed a moderate positive correlation (R^2^ = 0.614) between BCG- and ECG-derived respiratory rates, with a slope of 0.55, indicating a weaker relationship to that of heart rate. As shown in (Figure 3).

Figure 3 Comparison of RR between the ECG and BCG in awake dogs

### 3.2. Anesthetized Dogs’ Group 

The findings collectively indicate the accuracy, reliability, and clinical applicability of the BCG Sense1 wearable device as a viable non-invasive alternative for HR and RR monitoring in anesthetized dogs. As shown in (Figure 4)

Figure 4 Comparison of filtered BCG signals with ECG signals in anesthetized dogs during HR monitoring.

#### 3.2.1. Heart Rate

Under anesthesia, HR measurements showed a mean bias of −0.2 bpm and even wider limits of agreement (LOA) (−7.31 to 7.11 bpm), which were most likely caused by anesthesia-initiated physiological changes. Despite this, linear regression (R^2^ = 0.9151) remained consistently predictive, suggesting that the device holds the potential to be reliable for use during the anesthetized state As shown in (Figure 5).

Figure 5 Comparison of HR measurements between the ECG and BCG in anesthesia dogs.

#### 3.2.2. Respiratory Rate

The mean bias for RR was 0.79 bpm with an LOA of (−4.167 to 5.748 bpm), demonstrating close agreement between both methods. Linear regression (R^2^ = 0.7215) demonstrated a moderate correlation and confirmed the validity of RR monitoring under anesthesia with the BCG Sense1 device. As shown in (Figure 6).

Figure 6 Comparison of RR measurements between the ECG and BCG in anesthesia dogs.

## 4. Discussion

This study evaluated the clinical utility of the Sense1 wearable device based on the BCG system for dogs’ vital sign monitoring, comparing its performance against the gold-standard ECG method under both awake and anesthetized conditions. The primary objective was to assess the accuracy and reliability of BCG in measuring HR and RR across varying physiological states. Our findings strongly suggest that BCG offers a significant advancement in veterinary monitoring technology, providing a non-invasive, accurate, and efficient alternative to ECG.

Under awake conditions, Bland–Altman analysis and linear regression revealed robust correlations between BCG and ECG measurements for both HR and RR (HR: r = 0.97 and R^2^ = 0.94; RR: r = 0.78 and R^2^ = 0.61; *p* < 0.001) [19,20]. These high correlation coefficients, coupled with minimal bias and narrow 95% limits of agreement, demonstrate the high accuracy and reliability of BCG in measuring canine vital signs under typical physiological conditions [21,22]. The exceptionally strong correlation for HR (R^2^ = 0.94) is noteworthy, indicating that BCG effectively captures the vast majority of the variance in the ECG-measured HR. However, the slightly lower correlation coefficient for RR (R^2^ = 0.61) likely reflects the greater inherent physiological variability in respiratory patterns compared to the relatively consistent cardiac rhythm [23].

The study found significant differences in respiratory rate variability between awake and isoflurane-anesthetized dogs, emphasizing the need to consider the physiological state and assess the impact of physiological stress and anesthetic protocols.

The greater RR variability observed in awake dogs stems from the intricate interplay of autonomic nervous system (ANS) regulation, behavioral influences, and environmental factors [24,25]. External stimuli (environmental sounds, odors, and physical activity), emotional state (stress, excitement), and thermoregulatory mechanisms (panting) all significantly influence the RR [26,27]. The ANS, mediating both sympathetic and parasympathetic responses, plays a central role in this dynamic regulation. In addition, awake animals continuously adjust their RR in response to metabolic demands and maintain homeostasis in the face of transient hypoxia or hypercapnia. Sensory inputs from pulmonary stretch receptors and blood chemoreceptors further modulate respiratory depth and frequency [25,28]. In contrast to the dynamic RR patterns observed in awake dogs, anesthetized animals exhibit significantly reduced variability. This reflects the profound pharmacological effects of isoflurane, which directly depresses the brainstem respiratory center [29,30], resulting in a dose-dependent reduction in RR and a more regular breathing pattern driven primarily by passive gas exchange [31]. The type and depth of anesthesia further influence the extent of respiratory depression [32], leading to consistently lower RR variability in anesthetized animals compared to awake animals. Furthermore, anesthesia markedly blunts or eliminates the reflexive responses to metabolic demands and sensory inputs that normally modulate respiration in conscious animals [33,34].

Despite the inherent variability in RR, especially in awake animals, which is influenced by complex autonomic and behavioral factors, significant correlations between BCG and ECG measurements persisted under anesthesia (HR: r = 0.96 and R^2^ = 0.92; RR: r = 0.85 and R^2^ = 0.72; *p* < 0.01). These findings highlight the robustness and reliability of BCG, maintaining a strong correlation even under anesthesia despite the physiological disturbances associated with it, a challenge acknowledged in prior research on accurate respiratory rate measurement in anesthetized animals [35,36]. The ability of the BCG algorithm to effectively filter extreme values (upper and lower 20%) enhances its reliability, particularly under conditions of arrhythmia, where ECG accuracy can be compromised by the irregularity of R-wave signals. This characteristic could prove especially valuable in emergency and critical care settings, where rapid and precise vital sign assessment is crucial [37]. The ease of application and the non-invasive nature of the BCG Sense1 device offer significant advantages over ECG, which requires shaving, skin preparation, and electrode attachment, procedures that can be stressful for animals and may lead to erroneous measurements due to heightened physiological stress responses [38]. The lower recorded HR observed in dogs using BCG, compared to those measured using ECG, further suggests a reduction in stress levels using the BCG device [39]. The collection of 8000 high-quality data points, following rigorous data quality control, provided a robust dataset for this comparative analysis, strengthening our conclusions. This reduction in stress and the potential for early detection of subtle changes in vital signs [40,41] further enhance the clinical utility of this technology, particularly in clinical settings requiring continuous monitoring such as perioperative, intraoperative, and postoperative care [42].

The fact that high-quality data can be achieved continuously, even under difficult physiological states like anesthesia, renders BCG a useful technique for enhancing animal welfare and veterinary care for the patients [43,44].

## 5. Conclusions

This study has rigorously evaluated the BCG Sense1 wearable device, confirming its effectiveness in monitoring the heart rate and respiratory rate in dogs. Strong correlations with electrocardiographic (ECG) measurements were observed in both awake and anesthetized dogs. However, the study also revealed notable variability in respiratory rate (RR) readings, particularly among awake dogs, suggesting the need for further investigation into the influence of factors such as breed, anesthetic protocols, and underlying health conditions on BCG accuracy. Despite this limitation, the observed correlations, particularly the consistent accuracy of heart rate measurements, indicate that BCG technology holds significant promise for improving animal welfare and operational efficiency in veterinary care. Future research should focus on refining the methodology and expanding its application across diverse clinical populations to further elucidate its clinical utility.

## Figures and Tables

**Figure 1 vetsci-12-00301-f001:**
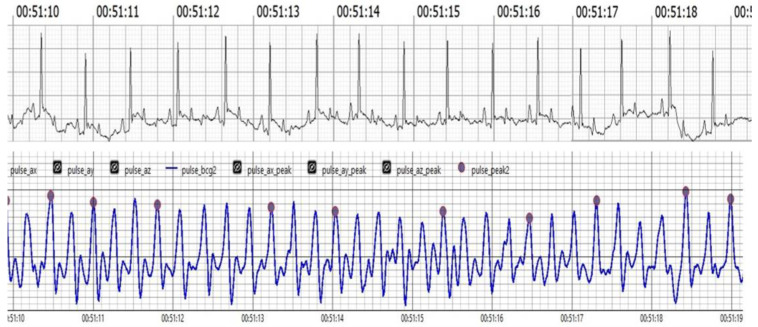
The upper trace illustrates the ECG, showcasing the electrical activity of the heart with clearly visible R waves. The lower trace displays the BCG, depicting mechanical oscillations associated with cardiovascular activity. Notably, the J peaks in the BCG trace appear analogous to the R waves in the ECG trace, demonstrating synchronization between the two measurement methods. This visual alignment underscores the reliability of BCG in capturing heart dynamics in real-time.

**Figure 2 vetsci-12-00301-f002:**
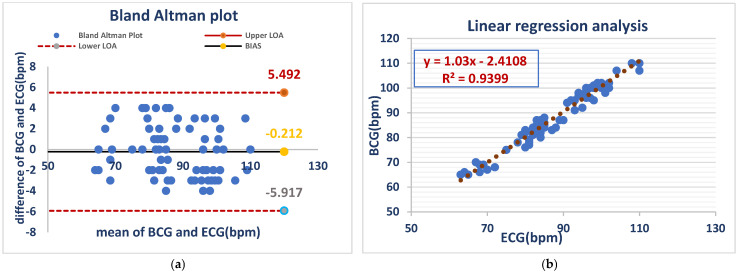
ECG and BCG measurements showed negligible bias (−0.212 bpm) but considerable variability (LOA: −5.917 to +5.492 bpm) (**a**). ECG and BCG HR measurements in awake dogs showed a strong positive linear correlation (R^2^ = 0.9399), with the BCG heart rate increasing by approximately 1.03 bpm for every 1 bpm increase in the ECG heart rate. This indicates high agreement between the two methods in awake dogs (**b**).

**Figure 3 vetsci-12-00301-f003:**
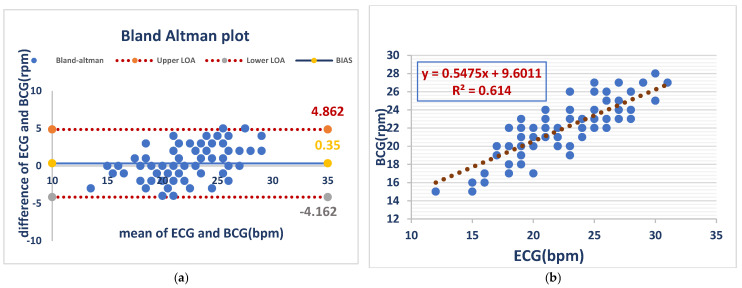
RR measurements from the BCG and ECG in awake dogs showed minimal bias (0.35 bpm) but notable variability (LOA: −4.2 to 4.9 bpm), indicating acceptable average agreement but inconsistent individual readings (**a**). BCG and ECG respiratory rates showed a moderate positive correlation (R^2^ = 0.614), with a slope of 0.55. This indicates a weaker relationship than that observed for heart rate (**b**).

**Figure 4 vetsci-12-00301-f004:**
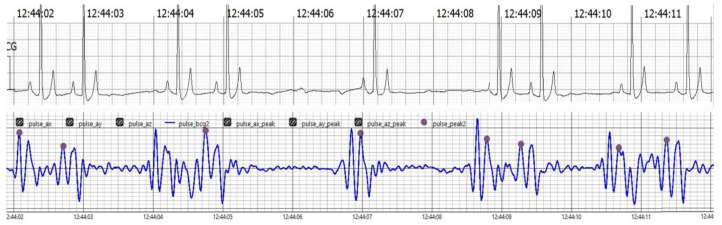
The upper trace displays the ECG, showcasing clear R waves indicative of the heart’s electrical activity. The lower trace presents the BCG, which captures mechanical oscillations related to cardiovascular activity. Notably, J peaks identified in the BCG signal, marked in pink, correspond with the R waves in the ECG trace, affirming the BCG’s effectiveness in tracking heart dynamics alongside the ECG.

**Figure 5 vetsci-12-00301-f005:**
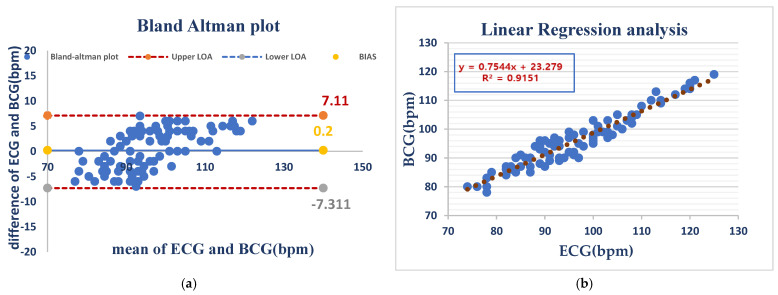
ECG and BCG heart rate measurements in anesthetized dogs showed negligible bias (0.2 bpm) but considerable variability (LOA: −7.3 to +7.1 bpm), indicating good average agreement but inconsistent individual readings (**a**). BCG and ECG heart rates showed a strong positive correlation (R^2^ = 0.9151), indicating that BCG is a good predictor of the ECG-measured heart rate (**b**).

**Figure 6 vetsci-12-00301-f006:**
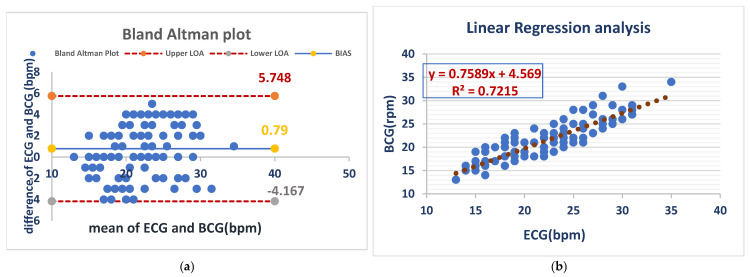
BCG and ECG heart rate measurements showed good agreement (bias = 0.79 bpm) but with considerable variability (LOA: −4.2 to +5.7 bpm). BCG slightly overestimated the heart rate compared to ECG (**a**). The BCG respiratory rate showed a moderate correlation with ECG (R^2^ = 0.72), with a slope of 0.76. This suggests that BCG is a less precise measure of respiratory rate than ECG (**b**).

**Table 1 vetsci-12-00301-t001:** Signalments of dogs included in the study.

Number	Breed	Age (Year)	Body Weight (kg)	Gender
1	Beagle	3.8	14.1	Male
2	Beagle	3.8	11.2	Male
3	Beagle	3.8	13.4	Male
4	Beagle	3.8	11.6	Female
5	Beagle	3.0	12.8	Female
6	Beagle	3.0	11.5	Female
7	Mixed	1.0	22.3	Male
8	Mixed	1.2	26.2	Male
9	Mixed	6.5	21.7	Male
10	Mixed	7.3	31.2	Male
11	Mixed	1.6	19.4	Female
12	Mixed	10.2	24.3	Female

## Data Availability

The data are contained within the article.

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
