# Peer review of "Clinical Application of Monitoring Vital Signs in Dogs Through Ballistocardiography (BCG)"

_vetsci, 2025, doi:10.3390/vetsci12040301_

Round 1
Reviewer 1 Report
Comments and Suggestions for Authors
This is a well-designed study that can be replicated. The results will be of interest for the practice.
Detailed comments
Page 1
Line 16: Write out BCG first.
Line 44: dog is already in the title; replace for anesthesia
Page 2
Lines 86 and 96: What is meant with the term “guardians”? Owners? If so, the please replace.
Line 93: How were the dogs exercised? How much time after exercise was the electrocardiogram measured?
Line 88: Replace “for” for “with”
Page 3
Line 108: Write out MEMS
Line 132: Mean and standard deviation (or error) should be added because results are presented, albeit in the discussion only. The results could be described in the results section already.
Page 4
Line 154: Lower case for all words but BCG and Comparison. Same for all headings of figures.
Line 160: “Included” with lower case
Author Response
I uploaded the file

Reviewer 2 Report
Comments and Suggestions for Authors
- Title needs optimization: "Clinical application of monitoring vital signs in dogs through Ballistocardiog-raphy (BCG)".
- Is the sample size sufficient for the control group and experimental group, each with 6 dogs? Need to provide evidence and explanation.
- Abstract: Focus on the important findings of this study.
- Introduction: The last paragraph is too short, merge it with the third paragraph. line 75: Therefore, this study aims to....
- Materials and Methods: The secondary title needs to be numbered.
- Will the age difference of the dogs used in this study have an impact on the results? Please provide a detailed explanation.
- Results: The figures‘ quality and clarity need to be improved.
- Discussion: The discussion is relatively simple and needs to be further enriched in the discussion section.
- The content of the funding and Acknowledgments sections is completely repeated.
- References need to be appropriately added.
- The entire text needs to be checked and polished for writing and grammar.
The English could be improved to more clearly express the research.
Author Response
I uploaded the file. Thank you

Reviewer 3 Report
Comments and Suggestions for Authors
This study briefly reported the utility of BCG Sense1 wearable device and confirmed its effectiveness for monitoring vital signs of dogs in comparison with ECG. By recording the heart rate and respiratory rate of dogs and analyzing the correlation with the established ECG standard in both healthy and anesthetized dogs, the authors found that BCG technology is an easy and accurate non-invasive monitoring for dog surgery, which could be served as an essential tool for both clinical practice and veterinary research. Interestingly, the integration of BCG technology could result in a better approach to monitor vital signs of pet animal and raise the standard of veterinary care. This innovation is meaningful for the welfare of dogs, and therefore it is acceptable for publication.
Author Response
I uploaded the file. Thank you

Reviewer 4 Report
Comments and Suggestions for Authors
This manuscript is about testing and proving the utility of vital sign monitoring via Ballistocardiography in dogs
The manuscript is well written and presented
However, I have some remarks
The authors should add more references in the discussions
The authors should add perspectives in the conclusion
Author Response
I uploaded the file. Thank you

Reviewer 5 Report
Comments and Suggestions for Authors
The authors aim to describe the use of a novel approach to monitor the cardiac activity in awake and anesthetized dogs. the scientific approach to the aim is well describer and appropriate, providing a comparison with the ECG which is considered a gold standard.
the purpose of the study identifies a gap between conventional diagnostic tool and recent advancemen in more technologically monitoring system.
however I would suggest to improve the materials and methods /results section:
- how long the measurements were taken for?
- the awake dogs were standing still, lying down (which recumbency), were they constrained some how?
- please define the type and duration of the exercise for the awake dogs
- is there any difference of measurement between the awake dogs at rest and after exercise ?
- I would replace the word healthy with "awake", because the anesthetized dogs are not sick (healthy status was an inclusion criteria also for this group, as said by the authors).
- I would also replace guardians with "owners"
- line 96-97: I think that a verb is missing in the sentence. please check
- line 100:"All dogs were healthy", this was already stated at line 98
- line 112-115: this is not a comment to be added here. I would rather move I at the discussion or introduction as a presentation of the device
- how was reparatory rate monitored as a gold standard? ECG doesn't monitor RR
- please define which recumbency was used during anesthesia.
- line 138-139: this is not objectively measured. It is a secondary Evaluation, derived from the numerical measures.
- in fig 1 and 4 please highlight the J peaks and the R waves in the graphs.
- line 178-179: this are comments that are true but should be moved to the discussion
- line 180: "anesthetic state"... this is the paragraph belonging to the awake dogs, isn't it? please check
- fig 3 legend (from line 191-196): it is a repetition
- BCG device is a wearable device that should lie close to the carotid artery, if I am not wrong. Was is fixed still somehow to the neck or left it free to move ?
- conclusion: the results surely suggest the device has some potentials for remote monitoring of the healthy status of awake and anesthetized dogs. However the study carries some limitations (some of them might be overtaken after the review) that should make the authors not so rigorous in the conclusion.
- line 304-30'5: please delete from "fof" to "used"
- line 312 - 313: please remove from "Please" to "reported"
- data availability statement: the journal instructions must be deleted
Author Response
I uploaded the file. Thank you

Round 2
Reviewer 2 Report
Comments and Suggestions for Authors
Accept in present form.
Author Response
Thank you for your feedback. The discussion section was revised substantially to address identified shortcomings and to enhance its clarity, depth, and analytical rigor. In response to reviewer comments highlighting shortcomings in the discussion section, this section has undergone a complete revision. The revised discussion now provides our study improvements, e.g., a more thorough analysis of the results, a more comprehensive review of the relevant literature, and a more detailed discussion of the study's limitations and implications.

Reviewer 5 Report
Comments and Suggestions for Authors
I thank the authors for the detailed response. However I would prefer that some of the data given in the response letter would be added in the main text.
I strongly suggest the authors to be more accurate in preparing the review, since I could find many little mistakes and confusion in the last version of the manuscript compared to the first one.
here you can find some suggestion.
line 103-104. it is still not clear how was each measurment after exercise taken for.
Check spelling, grammar and spaces between words throughout the text
line 186: the dogs are all healthy... which group belongs the this graph to? and the previous one? please check the legends
no mention to RR is made in the method section. but the data is described in the results
response 3: I meant to know if the agreement between the two methods was different between pre and post exercise.
You have to specify in the text the duration of the measurments
Response 8: you have to add this observation about RR in the text
response 9: can you add this in the text?
line 135-144: these are all results. you have to add here a description of the statistical analysis but not its results
response 10: I meant that the "The clinical utility of BCG was compared and evaluated compared to ECG under anesthesia that can affect cardiac activity."I know that the strength of correlation is objectively measurable, even though (as I said before) this is not the right place to describe the results.
response 17: so if the animal is moving this is not so realiable? please clarify
Author Response

(The authors gave the same response as above.)

Round 3
Reviewer 5 Report
Comments and Suggestions for Authors
I thank the authors to the attention given to my suggestion I found the paper more complete. It is suitable for pubblishing now.